# Noncommunicable disease behavioral risk factors in Sub Saharan Africa: A protocol of systematic review and meta-analysis

**Assefa Andargie Kassa**[1]*, **Segenet Zewdie**[2], **Mekuanint Taddele**[1]

**1** Department of Public Health, College of Medicine and Health Science, Injibara University, Injibara, Ethiopia, **2** Department of Pharmacy, College of Medicine and Health Science, Injibara University, Injibara, Ethiopia

* assefaand@gmail.com

**Data Availability Statement:** Deidentified research data will be made publicly available when the study is completed and published.

## Abstract

### Background

The most frequent risks of noncommunicable diseases include tobacco use, harmful use of alcohol, unhealthy diet, and physical inactivity. In low-income countries, it is not fully understood how serious these risk factors are. To address the issue at the risk factor level, it is essential to produce evidence that aids in the development of policies and initiatives in the area. This review is aimed to estimate the pooled prevalence of noncommunicable disease behavioral risk factors in Sub-Saharan Africa.

### Methods

Studies published between 2016 and 2023 will be located using searches of the electronic databases PubMed, CINAHL, and African Index Medicus as well as Google and Google Scholar. Two authors will independently review the records, and information will be taken from studies that present statistics on the prevalence of tobacco use, alcohol use, unhealthy diet, and insufficient physical activity among people older than 18 years. Using the $I^2$ and Q statistics, heterogeneity between studies will be evaluated, and it will be investigated using subgroup analyses and meta-regressions. Random effects meta-analysis model will be used and subgroup analysis will be performed by country, study design, and study year.

### Discussion

The burden of noncommunicable disease risk factors varies throughout the Sub-Saharan Africa. The review will be essential for both research and policy. The finding may even help to identify settings or subgroups of the population where noncommunicable diseases is of higher concern and help to set prevention priorities, to optimize resource allocation, and guide future research to fill knowledge gaps. The protocol has been registered in PROSPERO (CRD42023431348).

**Funding:** The author(s) received no specific funding for this work.

**Competing interests:** The authors have declared that no competing interests exist.

**Abbreviations:** AIDS, Acquired immunodeficiency syndrome; GRADE, Grading of Recommendations, Assessment, Development, and Evaluations; HIV, Human immunodeficiency virus; JBI, Joanna Briggs Institute; NCD, Noncommunicable disease; PROSPERO, Prospective Register of Systematic Reviews; SSA, Sub-Saharan Africa; WHO, World Health Organization.

## Introduction

Noncommunicable diseases (NCDs) are a huge public health and socioeconomic issue, accounting for around 41 million deaths worldwide equivalent to 74% of all deaths worldwide each year, with cardiovascular illnesses, cancers, chronic respiratory diseases, and diabetes accounting for more than 80% of all premature NCD deaths. The global mortality rate from one of the four major noncommunicable diseases is 17.8%. The NCD epidemic is wreaking havoc on individuals, families, and communities, and it threatens to overwhelm healthcare systems [1–3]. Millions of people have poor health conditions worldwide each year as a result of NCDs. The prevention, diagnosis, and treatment of NCDs cost the world millions of dollars each year. In its 2012 summit on sustainable development, the United Nations (UN) highlighted NCDs as one of the main obstacles to sustainable development in the twenty-first century [4].

The burden of NCDs is greater in developing countries [5]. According to estimates, there is a 20.8% chance of dying from major NCDs in African nations [3]. In fact, the majority of fatalities from NCDs—roughly three-quarters—occur in low- and middle-income nations, where the majority of the world's population resides, compared to about one-quarter in high-income nations. It is significant to remember that 17 million people die from NCDs annually before the age of 70, with the bulk of these premature deaths (86%) occurring in low- and middle-income countries [3, 6–9].

Sub-Saharan Africa (SSA), traditionally burdened by infectious diseases, is now experiencing a rising prevalence of NCDs [10, 11], driven by shifts in behavioral risk factors such as unhealthy diets, physical inactivity, tobacco use, and harmful alcohol consumption [12]. These modifiable behaviors are well-established determinants of NCDs globally, but in the context of SSA, they are driven by rapid urbanization, globalization, economic development, and socio-cultural changes [13]. These transitions have altered lifestyles and increased access to processed foods, tobacco products, and alcohol, while reducing opportunities for physical activity [14]. As a result, many individuals in SSA now face heightened exposure to these behavioral risk factors, contributing to the increasing NCD burden. The double burden of disease—where communicable diseases remain prevalent while NCDs rise—poses unique challenges for healthcare systems already strained by limited resources [10, 11, 15, 16].

The impact of these behavioral risk factors is substantial, as they not only lead to the development of NCDs but also influence their progression, severity, and related complications. Tobacco use, for example, is a significant cause of lung cancer and chronic respiratory diseases [17, 18], while harmful alcohol use has been associated with liver disease, hypertension, and various cancers [19, 20]. Similarly, unhealthy dietary practices, characterized by high consumption of fats, sugars, and salt, contribute to obesity, hypertension, and diabetes [21, 22]. Physical inactivity, on the other hand, is a critical contributor to obesity, cardiovascular disease, and certain cancers [23–25]. Together, these risk factors create a complex interplay that contributes to the worsening NCD landscape in the region [7, 10, 11].

Behavioral risk factors for NCDs are increasingly documented in SSA, but the region presents unique challenges for accurate data collection and synthesis. Diverse cultural, socioeconomic, and environmental contexts shape health behaviors, making it difficult to generalize findings across countries. Additionally, underreporting, lack of infrastructure, and limited access to healthcare further hinder comprehensive data collection on these risk factors. Despite these challenges, a growing body of literature on NCD behavioral risk factors in SSA has begun to emerge. However, these studies often vary in methodology, sample size, and geographic focus, creating inconsistencies in reported prevalence rates and associated health outcomes. Therefore, synthesizing the available evidence through a systematic review and meta-

analysis can provide a more accurate and comprehensive understanding of NCD risk factors in the region.

The proposed systematic review and meta-analysis will focus on behavioral risk factors for NCDs in SSA, consolidating evidence from studies across multiple countries. The primary objectives are to estimate the prevalence of key behavioral risk factors, including tobacco use, alcohol consumption, physical inactivity, and poor dietary habits among adults in SSA. By pooling data from different studies, this review aims to provide robust estimates that can inform public health policy and intervention strategies tailored to the region's unique contexts.

Given the rapid rise of NCDs in SSA, timely and effective interventions are essential. This review will serve as a critical resource for policymakers, public health officials, and researchers by providing evidence-based insights into the scope and scale of NCD risk factors. Moreover, understanding regional patterns of behavioral risk factors can help in developing targeted prevention programs that address the specific needs of SSA populations. Ultimately, this study will contribute to the global effort to reduce the burden of NCDs by highlighting the importance of behavioral risk factor reduction in SSA, a region at the intersection of traditional health challenges and emerging chronic disease threats.

## Materials and methods

### Protocol registration and review reporting

To prevent duplications, databases were initially searched for similar systematic reviews. We checked PubMed, Cochrane/Wiley Library, and PROSPERO to see if there has ever been a systematic review or meta-analysis on the subject. This protocol is prepared based on the Preferred reporting items for systematic review and meta-analysis protocols (PRISMA-P) 2015 statement [26] (S1 File) and the report will be carried out in compliance with the Preferred Reporting Items for Systematic Reviews and Meta-Analyses updated in 2020 (PRISMA 2020 Statement) [27]. The protocol has been registered in PROSPERO with registration number CRD42023431348.

### Patient and public involvement

No patients were involved in the design of this study since it is a review of primary studies. The findings of this systematic review will be presented at conferences or published in peer-reviewed journals.

### Eligibility criteria

The condition, context, and population (CoCoPop) components of the review questions are used to specify the eligibility criteria for studies on prevalence and incidence [28].

### Condition

The condition that we planned to address is behavioral risk factors of common noncommunicable diseases. These risk factors include tobacco use, alcohol use, unhealthy diet and insufficient physical activity.

**Tobacco use** is usage of any tobacco product, whether smoked or smokeless. Current use of tobacco is the use of one or more smoked or smokeless tobacco products on a daily or non-daily basis [29]. Studies reporting tobacco use using self-reported surveys including questions about current smoking status, frequency, duration, and amount of tobacco use or

Standardized questionnaires like the Global Adult Tobacco Survey (GATS) or Behavioral Risk Factor Surveillance System (BRFSS) will be included.

**Harmful use of alcohol** is drinking the amount of alcohol that causes detrimental health and social consequences for the drinker, the people around the drinker and society at large, as well as patterns of drinking that are associated with increased risk of adverse health outcomes [30]. Studies reporting harmful use of alcohol using Self-Reported Surveys (frequency, quantity, and type of alcohol consumed) or tools like the Alcohol Use Disorders Identification Test (AUDIT) or Cut-down, Annoyed, Guilty and Eye-opener (CAGE) Questionnaire will be considered eligible.

**Unhealthy diet** is defined as one with a low intake of fruits and vegetables and a high intake of salt [31]. Low fruit and vegetable intake refers to a daily intake of less than five servings (400 grams) of fruit and vegetables combined [30]. High salt consumption is defined as consuming more than 5 grams of salt or 2 grams of sodium per day [32]. Studies reporting the outcome using Self-Reported Surveys (Food frequency questionnaires (FFQs), 24-hour dietary recalls, or dietary diaries) or Standardized instruments like the Food Frequency Questionnaire (FFQ) or the 24-Hour Dietary Recall method will be included in the review.

**Insufficient physical activity** is defined as 150 minutes or less of moderate-intensity activity per week [33]. Studies reporting outcomes using Self-Reported Surveys (Questionnaires about the frequency, duration, and intensity of physical activity), tools like the International Physical Activity Questionnaire (IPAQ) or the Global Physical Activity Questionnaire (GPAQ) will be eligible. Studies that use incorrect or ambiguous outcome ascertainment criteria will be excluded.

## Context

This review will consider studies conducted in 49 countries of the SSA region. SSA refers to the region of the African continent south of the Sahara Desert. It comprises Central Africa, East Africa, Southern Africa, and West Africa. SSA contains a variety of geographical characteristics, including the Sahel region, savannahs, arid lowland terrain, and tropical forests [34]. While communicable disease and malnutrition are significant issues in the region, the burden of NCDs is increasing [35]. We will incorporate studies over the past ten years to acquire the most recent picture of NCD risk factors. The review will only include studies that have been published in English language.

## Population

Only research carried out among adult populations 18 years or older in SSA nations will be taken into consideration. Studies that only include children, specific populations like those with HIV, cancer, chronic respiratory diseases, and diabetes, studies that only cover one gender, and studies that focus solely on rural or urban populations will all be disregarded.

## Study design

The review will include designs aimed at determining the prevalence or incidence of behavioral risk factors for NCDs. Cohort studies and cross-sectional studies are two popular observational research approaches that should be taken into account in this study. The following data will be taken into consideration: 1) data from the general population (i.e., population prevalence surveys like the WHO STEPS survey), 2) data from local community-based studies with representative samples, and 3) data from screening programs.

## Information sources

To access published primary studies PubMed, African Index Medicus (AIM), and Cumulative Index to Nursing & Allied Health (CINAHL) database sources will be used. To supplement the electronic data base searches, the online archives of the WHO will be reviewed for applicable studies. Grey literatures will be retrieved using Google, Google Scholar and Institutional repositories. In addition, the reference lists of the retrieved studies will be probed to collect articles that are not accessible through databases as well as electronic search engines. During the search process, to suppress the number of irrelevant studies, the search will be restricted to only 'human studies' and 'English language' in the advanced search. The corresponding author(s) will be contacted via mail or other means of communication for articles with full texts that are hard to access.

## Search strategy

Studies will be identified in comprehensive search of all peer-reviewed published and unpublished studies that report the prevalence of NCD behavioral risk factors in SSA countries since 2016, after the official enforcement of the Sustainable Development Goals (SDG). Key search terms and keywords are identified through a preliminary search using the terms "Noncommunicable disease" AND "risk factors" AND "Sub-Saharan Africa". Final keywords and several corresponding search terms will then systematically be applied. In the advanced search of databases, the search strategy will be built based on the search terms using the 'Medical Subject Headings (MeSH)', title [ti], text word [tw] and [All fields] by linking them, with the "OR" and "AND" Boolean operators. Publication year (2016–2023), species (Human), Language (English), and age (19+ years) filters will be applied to make the search more specific. An example search strategy to be applied in the PubMed database is presented in the S2 File.

## Study records

The article search and screening activity will be done by AA, SZ. Articles searched from different sources will be exported to EndNote version 20.5, and duplicates will be identified and removed. The remaining articles will be evaluated in the context of the topic, study participants, language and study area. Irrelevant topics, studies conducted out of SSA countries and articles documented other than the English language will be rejected. The abstracts and full texts of the remaining studies will be reviewed. If the studies' full text cannot be accessed after an attempt to contact the original article author, they will be excluded.

## Data items

After selecting the appropriate research, two independent reviewers (AA and SZ) will use a prepared format on a Microsoft Excel spreadsheet to extract the pertinent data. Information such as the primary investigator's name, country, sample size, prevalence/proportion/incidence of NCD risk factors, response rate, study year, publication year, mean/median age of study participants, study design, proportion of male and female will be extracted. The prevalence, its logarithm, and its standard error (SE) will be calculated if the effect size is not reported in the primary studies. For any difficulties that might be encountered during data extraction, the corresponding author(s) will be contacted by e-mail.

## Risk of bias in individual studies

The quality assessment appraisal will be performed by two independent reviewers (AA and MT). The quality of each article will be assessed using the standardized Joanna Briggs Institute

(JBI) critical appraisal tool prepared for cohort studies, cross-sectional and analytical cross-sectional studies. All tools have 'Yes' or 'No' types of questions, and scores will be given 1 and 0 for 'Yes' and 'No' responses, respectively. Scores will be summed and transformed into a percentage. Only studies that scored ≥50% will be considered for both systematic review and meta-analysis of prevalence of tobacco use, alcohol use, unhealth diet and insufficient physical activity. When there are any scoring disagreements between the assessors, the sources of discrepancy will be investigated by a thorough discussion. For persistent disagreements in spite of the detailed review, a third independent reviewer (SZ) will be assigned as arbitrator. Moreover, the quality results of primary studies will be placed in a separate column of the data extraction format.

## Data synthesis

The extracted data will be exported to STATA Version17 for further analysis. The existence of heterogeneity among studies will be examined by $I^2$ heterogeneity test. The $I^2$ values of 25%, 50% and 75% will be interpreted as the presence of low, medium and high heterogeneity, respectively. Heterogeneity test ($I^2$) of ≥50% and a p-value of <0.05 will be declared as the presence of heterogeneity. If statistical heterogeneity is observed ($I^2 \geq 50\%$ or $P < 0.1$), the DerSimonian and Laird random-effects model will be employed [36]. Sensitivity analysis will be carried out to identify the influential studies that resulted in variation. Then, for extreme outlier studies, the extracted data will be checked for any error that might occur during the data extraction processes and if the data are free of errors, articles will be excluded from the analysis. Similarly, subgroup analyses will be employed by assuming the country, study design and the year of the study as grouping variables and sources of variation. In addition, meta-regression will be considered as a means to explain the existing heterogeneity. If quantitative synthesis is not appropriate, a systematic narrative synthesis will be provided with information presented in the text and tables to summarize and explain the characteristics and findings of the included studies.

## Meta-bias(es)

Publication bias will be detected by the funnel plot and Egger's regression test [37]. Accordingly, asymmetry of the funnel plot and/or statistical significance of Egger's regression test (p<0.05) will be suggestive of publication bias. Therefore, a non-parametric trim and fill (Duval and Tweedie's) method of analysis will be done [38]. Using the Dersimonian-Laird random-effects model, the pooled prevalence of NCD behavioral risk factors will be reported. The overall strength of the body of evidence will be assessed using the Grading of Recommendations, Assessment, Development, and Evaluations (GRADE) assessment tool. The GRADE framework is a systematic and transparent approach to grading the certainty of evidence in systematic reviews and clinical practice guidelines, as well as formulating and determining the strength of clinical practice recommendations [39]. The framework can be adapted and used for systematic reviews and meta-analyses of prevalence studies, although it was initially designed for intervention studies. When applying GRADE to prevalence data, the focus shifts from evaluating the effects of interventions to assessing the certainty of estimates related to the prevalence of a condition, risk factor, or outcome. This adaptation is often referred to as GRADE for prognostic or epidemiological studies [40].

## Ethics approval and consent to participate

This is a protocol for systematic review and meta-analysis that will synthesize data from previously published research. No new data will be collected from human participants or animals

for this review. Therefore, ethical approval and consent to participate will not be required for this study.

## Discussions

Although the burden of NCDs is increasing in SSA [10], data on NCDs and their risk factors is limited and not well organized [7]. The most significant modifiable risk factors, particularly the four behavioral risk factors for NCDs (tobacco use, unhealthy diets, alcohol abuse, and physical inactivity), which in turn result in the metabolic/biological risk factors, are the main focus of prevention and control for the major NCDs [6, 41]. There is clear evidence of a connection between behavioral risk factors and NCDs [42]. Taking early action to combat the NCD epidemic is strongly supported by the scientific evidence currently available [43]. Over a third of cancer-related deaths, as well as 80% of deaths from cardiovascular disease, and diabetes can be prevented by avoiding behavioral risk factors [1]. Adopting NCD prevention techniques is therefore thought to be the most economical, manageable, and long-lasting course of action to deal with the burden of NCDs globally [6].

Despite the efforts to control NCD risk factors, there are still gaps in ratification and enforcement of policies and strategies in SSA. The burden of NCD risk factors varies throughout SSA nations. This is the first systematic review and meta-analysis protocol that aims to estimate the combined prevalence of NCD risk factors in the SSA, to the best of our knowledge. We are confident that our data will be essential for both research and policy. The data may even help to identify settings or subgroups of the population where NCD is of higher concern and help to set prevention priorities, to optimize resource allocation, and guide future research to fill knowledge gaps. The data may even help to identify settings or subgroups of the population where NCD is of higher concern.

Due to the population variability in the SSA, this systematic review and meta-analysis may not be able to produce a reliable pooled estimate of NCD behavioral risk factors. The biases in the original studies that affect the outcome in the pooled estimate are another issue because the review is based on observational primary researches. Recall, social desirability, and observer biases are frequently encountered in observational studies. Variation in outcome measurement between the primary studies is another potential limitation in the review.

## Supporting information

**S1 File. PRISMA-P checklist.**
(DOCX)

**S2 File. Proposed search strategies for selected databases.**
(DOCX)

## Author Contributions

**Conceptualization:** Assefa Andargie Kassa.

**Methodology:** Assefa Andargie Kassa, Segenet Zewdie, Mekuanint Taddele.

**Project administration:** Assefa Andargie Kassa.

**Supervision:** Segenet Zewdie.

**Validation:** Assefa Andargie Kassa, Segenet Zewdie.

**Writing – original draft:** Assefa Andargie Kassa.

**Writing – review & editing:** Assefa Andargie Kassa, Segenet Zewdie, Mekuanint Taddele.

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
