## [Decision Letter · Decision Letter 0]

13 Aug 2024

PONE-D-23-40052Noncommunicable disease behavioral risk factors in Sub Saharan Africa: A protocol of systematic review and meta-analysisPLOS ONE

Dear Dr. Kassa,

Thank you for submitting your manuscript to PLOS ONE. After careful consideration, we feel that it has merit but does not fully meet PLOS ONE’s publication criteria as it currently stands. Therefore, we invite you to submit a revised version of the manuscript that addresses the points raised during the review process.

Conducting a systematic review and meta-analysis on noncommunicable diseases in the Sub-Saharan Africa region is undoubtedly a significant effort for both the scientific community and the public health services.

The topic addressed is extremely pertinent given the burden and the urgent need to fill knowledge gaps in the field, specially in lower-income countries.

Please ensure to identify the changes suggested by the reviewer in your text when submitting the revised manuscript.

Kindly submit your revised manuscript by Sep 27 2024 11:59PM. If you will need more time than this to complete your revisions, please reply to this message or contact the journal office at plosone@plos.org. Please include the following items when submitting your revised manuscript:

We look forward to receiving your revised manuscript.

Kind regards,

Sheila Rizzato Stopa

Academic Editor

PLOS ONE

2. Please amend the manuscript submission data (via Edit Submission) to include author Mekuanint Taddele.

Reviewers' comments:

Reviewer's Responses to Questions

**Comments to the Author**

1. Does the manuscript provide a valid rationale for the proposed study, with clearly identified and justified research questions?

Reviewer #1: Partly

2. Is the protocol technically sound and planned in a manner that will lead to a meaningful outcome and allow testing the stated hypotheses?

Reviewer #1: Yes

3. Is the methodology feasible and described in sufficient detail to allow the work to be replicable?

Reviewer #1: Yes

4. Have the authors described where all data underlying the findings will be made available when the study is complete?

Reviewer #1: Yes

5. Is the manuscript presented in an intelligible fashion and written in standard English?

Reviewer #1: Yes

6. Review Comments to the Author

General Recommendations

Please, in accordance with the PRISMA-P final checklist, provide the email addresses of all the authors of the protocol.

Abstract

- Methods: I suggest that the registration be incorporated into the text of the abstract rather than listed as a separate item.

- Keywords: "Behavioral risk factors" was not found in the MeSH terms.

Introduction

- In the first paragraph, the authors chose to discuss the insufficiency of evidence that noncommunicable diseases can be transmitted from person to person through infection, vectors, biological inheritance, or genetic transmission. While the statement of insufficient evidence may be a cautious way to express the lack of data proving any form of direct transmission, it is important to emphasize that there is a substantial body of evidence identifying and describing behavioral risk factors for noncommunicable diseases. This is the primary focus of the study, and I suggest prioritizing this aspect in the introduction. Therefore, I recommend revising this first paragraph.

- Adding a justification in the introduction about the choice of the Sub-Saharan Africa region could enhance the context and relevance of the study.

- Line 51: I suggest replacing the word "malignancies" with "cancer."

- Lines 71 and 72: Include examples of societal factors, which include intricate interactions between socioeconomic and environmental parameters.

- Considering that the study population will include only adults (aged 18 years or older), I recommend adding this specification to the objective, in line with PRISMA-P recommendations. For example: “Therefore, the objective of this study is to determine the pooled prevalence of noncommunicable disease behavioral risk factors among adults in Sub-Saharan Africa (SSA).”

Materials and Methods

- The authors chose to specify the review questions; however, a systematic review can also provide information about gaps in knowledge, thereby informing future research efforts. In other words, it is important to avoid arbitrary decisions regarding inclusion criteria and data extraction. Therefore, I suggest making it clear that the review may not fully address all the questions.

- Justification for the Use of CoCoPop: I recommend including the following addition to justify the use of CoCoPop: “The condition, context, and population (CoCoPop) components of the review questions are used to specify the eligibility criteria for studies on prevalence and incidence.”

- Condition: for the condition, I suggest adding strategies or justifications for including different methods for calculating indicators. For example, in various studies and countries, the calculation of indicators such as alcohol abuse or even fruit and vegetable consumption may differ. How will the authors address these biases across different studies? Providing this clarification can enable careful planning and anticipate potential issues.

- Line 154 and 155: the population was previously cited as being aged 18 years or older. Ensure consistency in the description.

- Line 208: describe the Grading of Recommendations, Assessment, Development, and Evaluations (GRADE) framework.

Discussion

- Line 212: the term used here was "alcohol abuse," but previous citations used "harmful." Ensure consistency in terminology throughout the manuscript.

---

## [Author Response · Author response to Decision Letter 0]

14 Sep 2024

Dear Editor and Reviewer,

We would like to thank you for the valuable feedback on our manuscript (Manuscript ID: PONE-D-23-40052). We appreciate the time and effort taken to review our work and are grateful for the constructive comments that have helped us improve the manuscript. Below, we provide detailed responses to each of the Editor and reviewer’s comments, and we have revised the manuscript accordingly.

Response to the Editor

Response: Thank you. We attempted to adhere to the journal style requirement.

2. Please amend the manuscript submission data (via Edit Submission) to include author Mekuanint Taddele.

Response: Thank you, we included Mekuanint Taddele 

Response: We amended and checked all references for completeness and correctness. We used the PLOS-one reference style consistently. We have checked that there are no papers that have been retracted. We included about 14 additional references in the revised manuscript to make it up to date and more comprehensive. 

Response to reviewer comments

General Recommendations

• Please, in accordance with the PRISMA-P final checklist, provide the email addresses of all the authors of the protocol.

Response: Thank you. We included the e-mail addresses of all authors per the recommendation (see page 1, line 16 & 17).

Abstract

• Methods: I suggest that the registration be incorporated into the text of the abstract rather than listed as a separate item.

Response: Thank you, we included as a text at the end of the abstract (separate item is removed).

• Keywords: "Behavioral risk factors" was not found in the MeSH terms.

Response: We divided the phrase “Behavioral Risk factors” in to “Behavior” and “Risk factors”.

Introduction

In the first paragraph, the authors chose to discuss the insufficiency of evidence that noncommunicable diseases can be transmitted from person to person through infection, vectors, biological inheritance, or genetic transmission. While the statement of insufficient evidence may be a cautious way to express the lack of data proving any form of direct transmission, it is important to emphasize that there is a substantial body of evidence identifying and describing behavioral risk factors for noncommunicable diseases. This is the primary focus of the study, and I suggest prioritizing this aspect in the introduction. Therefore, I recommend revising this first paragraph.

Response: We appreciate this comment and we revised this section by including the burden and impact of behavioral risk factors globally and regionally. (See the revised manuscript from pages 3-6 for details).

Adding a justification in the introduction about the choice of the Sub-Saharan Africa region could enhance the context and relevance of the study.

Response: We included justifications for the choice of SSA. We choose this region for its Rapidly Rising NCD Burden, High Prevalence of Behavioral Risk Factors, Lack of Region-Specific Data, and Policy and Intervention Relevance. This is incorporated in the last three paragraphs of the revised manuscript (Page 5 & 6).

Line 51: I suggest replacing the word "malignancies" with "cancer."

Response: Thank you for the detailed concern to improve our manuscript and we accepted the comment, and corrected accordingly.

Lines 71 and 72: Include examples of societal factors, which include intricate interactions between socioeconomic and environmental parameters.

Response: We appreciate this concern. Societal factors refer to the social, economic, and environmental conditions that shape the behaviors, health, and well-being of individuals and populations. These factors include socio-economic factors (income, education, employment, housing, social class, …), environmental factors (Physical environment, urbanization, climate change, health services and infrastructures), cultural and social networks (community support, cultural beliefs and practices, social cohesion, discrimination and social exclusion), Political and policy context (health policies, social policies, regulations, political stability). However, we removed the paragraph containing these concepts to make the background section focused on behavioral risk factors.

Considering that the study population will include only adults (aged 18 years or older), I recommend adding this specification to the objective, in line with PRISMA-P recommendations. For example: “Therefore, the objective of this study is to determine the pooled prevalence of noncommunicable disease behavioral risk factors among adults in Sub-Saharan Africa (SSA).”

Response: Thank you, we specified “adult population” in the objective statement.

Materials and Methods

The authors chose to specify the review questions; however, a systematic review can also provide information about gaps in knowledge, thereby informing future research efforts. In other words, it is important to avoid arbitrary decisions regarding inclusion criteria and data extraction. Therefore, I suggest making it clear that the review may not fully address all the questions.

Response: Thanks. We agree with this comment as these questions are included within the objective. we prefer to avoid explicit listing of review questions.

Justification for the Use of CoCoPop: I recommend including the following addition to justify the use of CoCoPop: “The condition, context, and population (CoCoPop) components of the review questions are used to specify the eligibility criteria for studies on prevalence and incidence.”

Response: We accepted this comment and corrected accordingly.

Condition: for the condition, I suggest adding strategies or justifications for including different methods for calculating indicators. For example, in various studies and countries, the calculation of indicators such as alcohol abuse or even fruit and vegetable consumption may differ. How will the authors address these biases across different studies? Providing this clarification can enable careful planning and anticipate potential issues.

Response: Thank you. We included tools that should be employed in the measurement of the outcome. We expect high potential heterogeneity across studies with respect to outcome measurements. We planned to accommodate all these variations based on the tools used to assess the outcomes. As far as studies used tools like self-reported surveys or standardized tools, we will include them in the review irrespective of the cultural and other socio-economic variations of the included population. To account such variations group specific analysis will be done in the meta-analysis.

Line 154 and 155: the population was previously cited as being aged 18 years or older. Ensure consistency in the description.

Response: Thank you. Corrected as “18 years or older”.

Line 208: describe the Grading of Recommendations, Assessment, Development, and Evaluations (GRADE) framework.

Response: We accepted the comment and included description how the GRADE framework could be applied in systematic reviews of prevalence.

Discussion

Line 212: the term used here was "alcohol abuse," but previous citations used "harmful." Ensure consistency in terminology throughout the manuscript.

Response: Thank you. Comment accepted and we used “harmful use of alcohol” consistently. 

Sincerely, 

Assefa Andargie Kassa (Corresponding author)

---

## [Editor Report · Decision Letter 1]

18 Sep 2024

Noncommunicable disease behavioral risk factors in Sub Saharan Africa: A protocol of systematic review and meta-analysis

PONE-D-23-40052R1

Dear Dr. Kassa,

We’re pleased to inform you that your manuscript has been judged scientifically suitable for publication and will be formally accepted for publication once it meets all outstanding technical requirements.

Kind regards,

Sheila Rizzato Stopa, PhD

Academic Editor

PLOS ONE
---

## [Editor Report · Acceptance letter]

29 Sep 2024

PONE-D-23-40052R1 

PLOS ONE

Dear Dr. Kassa, 

I'm pleased to inform you that your manuscript has been deemed suitable for publication in PLOS ONE. Congratulations! Your manuscript is now being handed over to our production team.

Kind regards, 

on behalf of

Dr. Sheila Rizzato Stopa 

Academic Editor

PLOS ONE